# Roles of Oxidative Injury and Nitric Oxide System Derangements in Kawasaki Disease Pathogenesis: A Systematic Review

**DOI:** 10.3390/ijms242015450

**Published:** 2023-10-22

**Authors:** Mitsuru Tsuge, Kazuhiro Uda, Takahiro Eitoku, Naomi Matsumoto, Takashi Yorifuji, Hirokazu Tsukahara

**Affiliations:** 1Department of Pediatrics, Okayama University Academic Field of Medicine, Dentistry, and Pharmaceutical Sciences, Okayama 700-8558, Japan; uda-kazuhiro@okayama-u.ac.jp (K.U.); tsukah-h@cc.okayama-u.ac.jp (H.T.); 2Department of Pediatrics, Kawasaki Medical School, Kurashiki 701-0192, Japan; eitoku@med.kawasaki-m.ac.jp; 3Department of Epidemiology, Okayama University Academic Field of Medicine, Dentistry, and Pharmaceutical Sciences, Okayama 700-8558, Japan; naomim@okayama-u.ac.jp (N.M.); yorichan@md.okayama-u.ac.jp (T.Y.)

**Keywords:** biomarker, coronary artery lesions, Kawasaki disease, multisystem inflammatory syndrome in children, nitric oxide, nitrosative stress, oxidative stress, systemic inflammation, vascular endothelial dysfunction

## Abstract

Kawasaki disease (KD) is an acute febrile vasculitis that occurs mostly in children younger than five years. KD involves multiple intricately connected inflammatory reactions activated by a cytokine cascade. Despite therapeutic advances, coronary artery damage may develop in some patients, who will be at risk of clinical cardiovascular events and even sudden death. The etiology of KD remains unclear; however, it may involve both genetic and environmental factors leading to aberrant inflammatory responses. Given the young age of onset, prenatal or perinatal exposure may be etiologically relevant. Multisystem inflammatory syndrome in children, a post-infectious hyper-inflammatory disorder associated with severe acute respiratory syndrome coronavirus 2, has features that overlap with those of KD. Available evidence indicates that vascular endothelial dysfunction is a critical step in the sequence of events leading to the development of cardiovascular lesions in KD. Oxidative stress and the dysregulation of the nitric oxide (NO) system contribute to the pathogenesis of inflammatory responses related to this disease. This review provides current evidence and concepts highlighting the adverse effects of oxidative injury and NO system derangements on the initiation and progression of KD and potential therapeutic strategies for cardiovascular pathologies in affected children.

## 1. Introduction

Kawasaki disease (KD), previously known as mucocutaneous lymph node syndrome, was first reported by Dr. Tomisaku Kawasaki in 1967 [1,2]. KD is a systemic inflammatory disease complicated by medium-sized vasculitis found mostly in children younger than five years. Asian countries such as Japan, South Korea, China, and Taiwan exhibit higher rates of KD than the United States and Europe [3]. The incidence of KD per 100,000 children under five years of age is 330 in Japan [4], 134 in South Korea [5], 25 in the United States [6], and approximately 5–10 in Europe [3]. Unfortunately, the exact incidence of KD in Middle Eastern and South American countries remains unknown due to the lack of census data on KD.

In addition to the common symptoms of KD associated with systemic inflammation, 25–30% of patients develop coronary artery abnormalities if left untreated. After introducing high-dose intravenous immunoglobulin (IVIG) and additional therapies, coronary artery lesion (CAL) incidence decreased to 5% or less. Despite the therapeutic advances, KD is a leading cause of childhood-onset acquired heart disease in developed countries [6,7,8,9].

Using recent evidence, we explain the involvement of oxidative injury and nitric oxide (NO) system derangements in the development and progression of KD, including study findings of its biochemical and epidemiological characteristics, and we propose strategies for managing cardiovascular complications.

## 2. Clinical and Pathological Features of KD

Despite more than half a century passing since its initial report [1], the etiology of KD remains unknown. Epidemiological studies suggest that KD may develop in children with genetic predispositions when exposed to infectious agents or environmental triggers [6,7,8,9].

The diagnosis of KD is based exclusively on clinical findings [6]. The symptoms associated with KD include fever, bilateral bulbar conjunctival hyperemia, lip and oral mucosal changes, rash, peripheral extremity changes, and nonsuppurative cervical lymphadenopathy [3] (Figure 1). KD is diagnosed when five or more of the aforementioned six main symptoms are present during the course of the disease. The rash includes redness at the site of the Bacillus Calmette–Guérin injection scar and changes in the lip and oral mucosa, including lip flushing, strawberry tongue, and diffuse redness of the oropharyngeal mucosa. Changes in the peripheral extremities include hard edema of the hands and feet, erythema of the palmar soles or tips of the toes, and membranous desquamation of the fingertips. Laboratory tests are not part of the diagnostic criteria but include significantly elevated inflammatory markers, such as the erythrocyte sedimentation rate and C-reactive protein. Platelet counts may initially be normal at the time of diagnosis but typically increase by the second week of illness. If left untreated, clinical symptoms usually resolve within an average of 12 days.

Vasculitis resulting from KD can lead to coronary artery abnormalities, myocarditis, and myocardial infarction. CALs are the most common cardiovascular manifestation of KD, and small coronary artery disease is observed in 15–20% of children at the time of diagnosis [7]. Coronary artery rupture and myocarditis with arrhythmia may occur in the acute phase [10]. CAL progression may result in stenosis, occlusion, or thrombosis, leading to long-term cardiovascular complications, including ischemic heart disease [10]. Small aneurysms are more likely to regress and return to normal lumen diameters faster than large aneurysms. Approximately 25% of untreated children develop coronary aneurysms, and, despite IVIG treatment, approximately 5% of pediatric patients with KD develop coronary artery abnormalities [6,11]. The use of Z-scores allows for a more accurate evaluation of the severity of coronary artery dilation, correcting for body surface area [6,12]. According to Z-score classification, approximately 40% of KD patients in Japan will have initial Z-max scores ≥2.5, which are classified as coronary aneurysms. Large CALs increase the risk of myocardial infarction. Other long-term cardiovascular sequelae include electrocardiogram abnormalities, myocardial fibrosis, and systolic or diastolic dysfunction [13].

Although the clinical symptoms of KD are well characterized, its etiology has not yet been established. Several hypotheses have been proposed to explain this phenomenon. KD is thought to be an exaggerated immune response caused by genetic susceptibility, infectious diseases, and environmental triggers [6,7,8,9,14] (Figure 2). Single-nucleotide polymorphisms in various genes, such as ITPKC, CASP3, and FCGR2A, are associated with KD susceptibility, coronary aneurysm formation, and treatment resistance [15].

Moreover, epidemiological data on KD suggest that infectious diseases may be the cause; however, the causative source of the infection has not yet been clarified [16,17]. *Staphylococcus aureus*, streptococci, coronaviruses, enteroviruses, Epstein–Barr virus, and rhinoviruses are suspected to be associated with KD [18]. Furthermore, infection by pathogens and pro-inflammatory agents such as pathogen-associated molecular patterns (PAMPs), toxins (superantigens), and damage-associated molecular patterns (DAMPs) from injured or infected host cells can contribute to hyperinflammatory responses in KD. PAMPs in experimental KD mice include *Lactobacillus caseii* and *Candida albicans* [8]. DAMPs associated with KD include S100 proteins, high-mobility group box-1, heat-shock proteins, oxidized phospholipids, the apoptosis-associated speck-like protein containing a caspase recruitment domain, caspase-1, and gasdermin D [8]. Environmental triggers include air allergens, toxins, pollutants, and drugs, but no single factor has been identified [19,20].

These factors activate the innate and adaptive immune systems, causing an exaggerated inflammatory response. The innate immune system mainly increases interleukin (IL)-1 levels by activating the NLR family pyrin domain-containing 3 inflammasome. The adaptive immune system involves the activation of lymphocytes, extracellular trapping of neutrophils, and activation of macrophages [21]. As a result, inflammatory cytokines, such as IL-1, IL-2, IL-6, IL-8, interferon-α, and tumor necrosis factor-α, are produced, and multiple pro-inflammatory mediators are involved [8].

KD is classified as a medium-sized vasculitis characterized by the frequent invasion of arteries outside solid organs, such as the coronary arteries. The formation of coronary aneurysms due to coronary arteritis causes ischemic heart disease and significantly affects patient prognosis. Between six and eight days following disease onset, studies have reported the dehiscence of medial smooth muscle cells due to edematous changes within blood vessel walls, along with acute necrotizing coronary arteritis observed on the luminal and adventitial sides. A high degree of inflammatory cell infiltration, consisting mainly of neutrophils, monocytes, and macrophages, was observed in the intima and adventitia of coronary arteries [22]. Eight to ten days after disease onset, inflammatory cells rapidly infiltrated from the intima and adventitia toward the media, causing panarteritis that spread to all layers of the artery [23]. Infiltrating inflammatory cells produced and released various inflammatory cytokines such as IL-1β and tumor necrosis factor, proteolytic enzymes, and reactive oxygen species (ROS). These damaged structures, including the internal elastic lamina, medial smooth muscle, and intercellular matrix, play a crucial role in supporting the coronary artery’s structure and gradually contribute to the deterioration of the vascular walls. In addition, proliferative changes in fibroblasts and smooth muscle cells in the vascular lumen were observed, contributing to narrowing the vascular lumen [24]. Ten to twelve days after disease onset, the weakened coronary arteries expanded efferently, forming coronary lesions such as arterial dilation and aneurysms. Severe vascular wall inflammation persisted until approximately 25 days of illness and then gradually diminished, and inflammatory cells almost disappeared at around 40 days of illness [25].

## 3. KD and Multisystem Inflammatory Syndrome in Children

Coronavirus disease 2019 (COVID-19), caused by severe acute respiratory syndrome coronavirus 2 (SARS-CoV-2), was initially detected in Wuhan, China, in December 2019 and rapidly disseminated worldwide [26]. Clusters of children with Kawasaki-like syndrome were reported in European countries and the United States in April and May 2020 [27,28,29]. Subsequently, this condition was defined as “multisystem inflammatory syndrome in children” (MIS-C) by the Centers for Disease Control and Prevention (CDC) and the World Health Organization (WHO). With the COVID-19 pandemic, the emergence of MIS-C has attracted renewed attention and interest in KD.

MIS-C develops within 2–6 weeks of SARS-CoV-2 infection and presents with an acute systemic inflammatory syndrome involving the heart, lungs, kidneys, brain, skin, eyes, and gastrointestinal organs. KD and MIS-C share many clinical and pathological features. MIS-C tends to manifest in older patients and presents with a greater frequency of gastrointestinal and neurological symptoms, as well as cardiac involvement, including myocarditis, than classical KD (Figure 3). 

The prevalence of KD is consistently higher in males, with the male-to-female ratio reported to be 1.5 in the United States, 1.31 in Japan, and 1.62 in Taiwan [6,30]. Furthermore, male gender was reported to be one of the risk factors for complications of CALs in KD patients [31]. However, the sex ratio of MIS-C patients is slightly higher in males [32], and several studies reported no significant difference in the sex ratio between MIS-C and KD patients [33,34,35]. In addition, coronary artery abnormalities were more common in male patients than in female patients [36].

The emerging spread of MIS-C has been observed worldwide along with the COVID-19 pandemic, whereas the number of KD patients decreased by approximately one-third in a cohort study about KD incidence in Japan [37], suggesting a potential KD pathogenesis involving transmission among children. In the study of KD incidence in the United States between 2018 and 2020, the national and local (San Diego region) reduction in KD cases was associated with a period of school closures, masking mandates, decreased ambient pollution, and the decreased circulation of respiratory viruses, which all overlapped to different extents with the period of decreased KD cases. KD in San Diego rebounded in the spring of 2021, coincident with the lifting of the mask mandates [38]. These findings provide evidence of an association between social behavior and exposure to the agent(s) implicated in KD, consistent with the notion of a respiratory mode of entry for the agent(s). 

The etiology of KD remains complex and multifactorial. Genetic susceptibility and infectious triggers may also contribute to disease development. The association between KD and COVID-19 has been the subject of considerable debate [39]. A possible association between KD and a strain of human coronavirus, NL63, was reported in 2005 [40]. Subsequently, several studies, including those on other strains (229E, OC43, and HKU1), have been conducted; however, no solid evidence has been established to date [39]. In patients with MIS-C, the condition is assumed to be a post-infectious immune-mediated response to SARS-CoV-2 involving a cytokine storm. Further research is required to elucidate the underlying mechanisms and identify specific triggers that could lead to improved diagnostic methods and targeted therapies for pediatric inflammatory conditions.

## 4. L-Arginine and Nitric Oxide Production in Physiological and Pathological Conditions

### 4.1. L-Arginine–Nitric Oxide (NO) System

NO, a free radical and weak oxidant, exerts multiple biological effects over a wide spectrum, from physiological regulatory functions to damaging alterations that contribute to the pathogenesis of diverse diseases [41,42,43]. NO is synthesized in all cell types through NO synthases, utilizing L-arginine and molecular oxygen. In this process, citrulline is concurrently generated. There are three distinct nitric oxide synthase (NOS) isoforms: neuronal NOS (NOS1), inducible NOS (NOS2), and endothelial NOS (NOS3). NOS1 and NOS3 are constitutively expressed as calcium-dependent enzymes that release NO transiently in response to physiological stimuli. In contrast, NOS2 is a calcium-insensitive enzyme induced by pro-inflammatory cytokines and microbial-associated products in many cell types, including vascular endothelial cells, smooth muscle cells, hepatocytes, monocytes, and macrophages.

NOS1-derived NO has been implicated in synaptic plasticity and is a neurotransmitter in the central and peripheral nervous systems. NO generated by the high-output NOS2 enzyme plays a critical role in host defense by exerting cytotoxic effects on bacteria, viruses, parasites, and tumor cells. NOS2-derived NO contributes to the pathophysiology of inflammatory diseases and is the predominant mediator of hypotension in patients with septic shock. Although healthy cells in the vasculature do not present significant levels of NOS2, several pathological processes show increased NOS2 activity in the blood vessels [42,43,44]. As NOS2 can generate higher NO levels than NOS3, NOS2 activation leads to excess NO and the impairment of vascular function. Excess NO limits the response of the blood vessels to vasodilators and decreases NO sensitivity.

Alternatively, NOS3 maintains vascular health. In addition to acting as a barrier between the intravascular and interstitial compartments, the vascular endothelium is a widely distributed organ responsible for the regulation of blood flow; the control of vascular permeability; angiogenic vascular remodeling; and metabolic, synthetic, antithrombogenic, and anti-inflammatory processes. NOS3-derived NO prevents inflammation by inhibiting the synthesis of inflammatory cytokines and chemokines; the expression of surface adhesion receptors; and the recruitment, infiltration, and activation of leukocytes within the vasculature [43,45,46]. Conversely, the loss of NO expression or activity causes “endothelial dysfunction”, which is characterized by impaired endothelium-dependent vasodilation, endothelial activation, apoptosis, endothelial barrier disruption, arterial stiffness, vessel wall thickening, and a prothrombotic and inflammatory state [43,44,47,48].

The NOS product citrulline is subsequently recycled back to arginine via the serial actions of arginosuccinate synthetase and lyase. “Citrulline-arginine recycling” mainly occurs in blood vessels and the kidneys. The vascular endothelium maintains NO synthesis through this pathway. Arginase, the last enzyme in the urea cycle, hydrolyzes arginine to polyamines and proline. Arginase I is mainly present in the liver, whereas arginase II is primarily in the mitochondria of cells outside the liver. Arginase II is induced in many cells, including in vascular endothelial cells, smooth muscle cells, hepatocytes, and macrophages in inflammatory pathology (“arginine steal phenomenon”). NOS and arginase, which share arginine as a substrate, control the functions of each other [43,46,49,50] (Figure 4).

### 4.2. Asymmetric Dimethylarginine (ADMA), an Endogenous NOS Inhibitor

ADMA, an endogenous NOS inhibitor, is found in the body. During protein synthesis, the arginine residue in the post-translational peptide is methylated by protein arginine methyltransferase (PRMT). Free ADMA, formed during proteolysis, is actively degraded by the intracellular enzyme dimethylarginine dimethylaminohydrolase (DDAH), which catalyzes the conversion of ADMA to citrulline and dimethylamine. Free ADMA can be transported in or out of cells by a cationic amino acid transporter. ADMA can be transported to various organs, such as the lungs, liver, kidneys, intestine, and blood vessels (Figure 5). Although DDAH-1 is present in many tissues that express NOS1, DDAH-2 is mainly present in vascular tissues that express NOS3 [45,46,51,52]. 

Since ADMA competes with L-arginine for NOS and cell transport, the bioavailability of NO depends on the balance between the two. ADMA inhibits NOS3 and decreases NO production in the endothelium of vessel walls. The elevation of ADMA levels inhibits NO synthesis by NOS3 and could even uncouple the enzyme (“NOS3 uncoupling”), consequently enhancing oxidative stress. Elevated ADMA levels have been detected in various vascular diseases, including hypertension, atherosclerosis, diabetes mellitus, renal insufficiency, hepatic failure, hypercholesterolemia, hypertriglyceridemia, and hyperhomocysteinemia [43,45,46,47,51,52]. The dysregulation of the balance between the L-arginine-NOS-NO and PRMT-ADMA-DDAH pathways is likely involved in the accumulation of ADMA in tissues and blood, thereby contributing to cardiovascular pathologies (Figure 5). 

## 5. Oxidative Stress with Relevance to the L-Arginine-NOS-NO Pathway

The expression of cellular physiological functions requires the tonic formation of ROS, which are indispensable for signaling, organelle function, energy production, and the processing of unnecessary cells and foreign matter [53,54].

Oxidative stress is defined as an imbalance between the systemic production of ROS and the ability of a biological system to detoxify ROS or repair the resulting damage readily. ROS are chemically reactive entities that contain oxygen, either as oxyradicals or non-radical species, such as superoxide (O_2_^−^•), hydroxyl (OH•), alkoxyl (RO•), and peroxyl (ROO•) radicals, and hydrogen peroxide (H_2_O_2_). All cell types, including neutrophils, eosinophils, monocytes, macrophages, cytotoxic lymphocytes, epithelial cells, endothelial cells, and other resident cells, produce ROS; this is mediated by the activities of numerous enzymes. Excess ROS is strictly controlled by endogenous antioxidative defense mechanisms, which involve various enzymes, proteins, and low-molecular-weight molecules [45,46,55,56].

When ROS are formed excessively and beyond control in the body, they damage lipids, proteins, enzymes responsible for biological structure and function, and genomic DNA responsible for genetic information, harming tissues and organs and causing disease. Consequently, a vicious cycle ensues, in which oxidative stress is amplified, causing disease progression and damage to other organs. Although the body addresses increased oxidant production using the organ network, tissue damage advances when the stress is severe, prolonged, or incessant. “Enhanced oxidative stress” or “redox control failure” is involved in the progression of many human diseases [46,55,56,57,58].

NO formation and oxidative stress are interrelated. The L-arginine–NO system sustains a low steady-state level of ROS that are persistently produced in vivo. It achieves this by regulating ROS availability and cellular redox states, thereby modulating various cellular functions, including nuclear gene regulation [43,44,45,46,47]. Rats receiving chronic NOS inhibitor (nitro-L-arginine methyl ester) treatment exhibited endothelial dysfunction syndrome and various parenchymal lesions. Enhanced oxidative stress likely contributes to endothelial dysfunction and organ damage in chronically NO-depleted animals [59,60,61,62,63].

Chronic NO depletion likely causes increased oxidant production. NO provides cells with a reducing environment and scavenges O_2_^−^• formed during metabolism. The balance between NO and O_2_^−^• is a key determinant in promoting oxidative stress [43,44,45,46,47]. Under normal physiological conditions, endogenous NO inhibits oxidative reactions when NO is produced in greater amounts than O_2_^−^•. When the flux of each radical is identical, the maximum oxidant production occurs, presumably through the formation of peroxynitrite (ONOO^−^) and its conjugate acid ONOOH [43,44,64]. Diminished NO synthesis via chronic NOS inhibitor administration may produce a situation in which the balance between NO and O_2_^−^• in favor of NO is altered in favor of a more oxidative environment. It is also highly likely that chronic NO depletion leads to NOS3 uncoupling, which would enhance oxidative stress [43,44,45,46,47].

During the acute systemic inflammatory response, the large amounts of NO formed by NOS2 surpass the physiological amounts of NO usually formed by NOS1 or NOS3. Excessive NO produced by NOS2 reacts with ROS concomitantly formed by nicotinamide adenine dinucleotide phosphate (NADPH) oxidase and other sources, leading to the formation of reactive nitrogen species (RNS), predominantly ONOO^−^ [43,44,64]. Both ROS and RNS contribute to the host acute inflammatory response, either via direct cellular toxicity or through the activation of pro-inflammatory gene expression pathways [43,44,45,46,64,65]. 

## 6. Biomarkers for Oxidative Stress and NO System

As described in the previous section, ROS and RNS directly impact the structure and functionality of numerous enzymes, proteins, lipids, and DNA through oxidation and nitration reactions. Any biomolecule can be damaged by ROS (including ONOO^−^) [43,44,64,65]. The primary cellular target of oxidative stress depends on the cell type, characteristics of the imposed stress, generation site, proximity of the ROS to a specific target, and stress severity. Typically, because the half-life of ROS is short, special techniques must be used to detect ROS in vivo. “Oxidative stress biomarkers” can determine the extent of oxidative injury and indicate the oxidant source. They are important for predicting the consequences of oxidation and providing a basis for designing appropriate interventions to alleviate or prevent injuries. Briefly, oxidative stress biomarkers are separated into two categories: the formation of ROS-modified molecules (including ONOO^−^) and the consumption or induction of enzymes or antioxidants. The measurement of these biomarkers in body fluids enables repeated monitoring of the oxidative stress status in vivo, which cannot be performed with invasive tests [45,46,57,66].

The criteria for considering a biomarker of oxidative stress include its minimal accumulation, resistance to metabolism, and stability within the body. The emerging availability of enzyme-linked immunosorbent assays for oxidative stress biomarkers allows for their application to assess various pathological conditions [45,66]. Recently, there have been significant advancements in the development of instruments that allow for the rapid measurement of biomarkers associated with oxidative stress. These instruments can assess blood total hydroperoxides (THs), the biological antioxidative potential (BAP), urinary 8-hydroxy-2-deoxyguanosine, and the L-type fatty acid-binding protein using very small sample sizes, typically within just 10 min. This progress has made it possible to evaluate the oxidative stress status at the patient’s bedside, enhancing the efficiency of clinical assessments [67].

The important advantages of biomarker measurement in clinical practice are as follows: (1) the therapeutic effects of “antioxidants” can be evaluated noninvasively and minimally invasively (for a patient-friendly continuous evaluation); (2) the pathology can be analyzed from the viewpoint of the biological response (for rational multidisciplinary treatment); (3) similar markers can be measured in model animals (for optimal translational research); and (4) rapid tests are applicable (for proper judgment at an early stage) [46].

The clinical utilization of “antioxidants” in treating diseases necessitates several prerequisites: (1) the drug demonstrates antioxidative properties at the cellular, tissue, or organ level; (2) oxidative stress plays a role in the development or progression of the disease; (3) the drug shows efficacy in disease models; (4) there is a history of the drug’s use in humans; and (5) clinical trials involving human participants have been conducted [46]. Well-known drugs that meet these five requirements and are used clinically include edaravone in the brain, hydroxymethylglutaryl (HMG)-coenzyme A reductase inhibitors in the cardiovascular field, and angiotensin-converting enzyme and angiotensin 1 receptor inhibitors in the kidney. The measurement of oxidative stress biomarkers is translational. Figure 6 summarizes the representative oxidative stress biomarkers and their clinical significance.

## 7. Oxidative Stress Status in the Acute Stage of KD

During the acute phase of KD, the majority of cells infiltrating the coronary arteries are neutrophils, monocytes, and macrophages, which are primary sources of ROS (including ONOO^−^) [6,7,8,9]. ROS extend the damage to the inflammatory cells themselves and adjacent cells, such as vascular endothelial and smooth muscle cells [43,44,64,65]. Activated neutrophils, monocytes, and macrophages also release large amounts of myeloperoxidase, a pro-oxidant enzyme that amplifies the formation of ROS (including ONOO^−^) and the development of coronary arteritis in KD.

Therefore, it is easy to predict that oxidative and nitrosative stresses are intensively enhanced in the acute stage of KD [8,9,58,68]. Herein, we refer to leading articles to determine the pathological roles of oxidative and nitrosative stresses in the acute stage of KD. 

Uchida et al. [69] reported heightened levels of lipid peroxides in plasma, as well as increased manganese superoxide dismutase in polymorphs and lymphocytes, along with elevated glutathione peroxidase and catalase in erythrocytes during the acute phase of KD. In contrast, copper/zinc superoxide dismutase levels in the polymorphs, lymphocytes, and erythrocytes were unaltered. Acute infectious diseases did not alter the blood’s lipid peroxides or antioxidant enzyme levels. Lebranchu et al. [70] also reported that lipid peroxidation products (malonyldialdehyde) and organic THs increased, and zinc levels decreased in the plasma during the acute stage of KD; however, these levels returned to normal after IVIG treatment.

Yahata et al. [71] simultaneously measured blood TH and BAP levels using a free radical elective evaluator (FREE^®^; Wismerll Co. Ltd., Tokyo, Japan) [45,67]. This study included 19 patients. They were classified as responding well (Group A) or poorly (Group B) to IVIG (2 g/kg, single dose). Initially, both groups had significantly higher blood TH levels than healthy controls. Blood TH levels significantly decreased in Group A immediately and two weeks after IVIG treatment but did not decrease in Group B until two weeks post-treatment. BAP levels were unremarkable in Group A, but they were significantly lower in Group B than in the other groups. BAP significantly increased in Group A two weeks after IVIG treatment but remained low in Group B. The study results indicated that acute-stage KD patients present with obvious hyperoxidant stress, which improves in response to IVIG treatment in most patients. Additionally, the potential utility of blood BAP as a predictive marker for IVIG treatment responsiveness is suggested. Kaneko et al. [72] reported similar results. Serum TH levels before IVIG administration were significantly higher in the KD group than in the bacterial infection (BI) group. The BAP levels before IVIG were lower in the KD group than in the BI group, although the difference was insignificant. The TH/BAP ratio (oxidative stress index) was significantly higher in the KD than that in the BI group. After IVIG treatment, blood TH levels decreased, and BAP levels increased. As a result, the “oxidative stress index” decreased significantly after IVIG in the KD patients.

Takatsuki et al. [73] found significantly elevated urinary 8-isoprostane levels in KD patients before IVIG administration compared to non-KD febrile patients (control). Patients with KD were administered three different regimens of IVIG (2 g/kg for one day, 1 g/kg for one day, and 400 mg/kg/day for five days). All IVIG regimens significantly reduced urinary 8-isoprostane levels seven days after administration.

Yachie et al. [74] found that heme oxygenase-1 mRNA expression and production in circulating monocytes were significantly elevated in patients with KD compared to those in the control group. These results suggested that monocyte heme oxygenase-1 is a potent anti-inflammatory agent that controls excessive cell or tissue injury in oxidative stress and hypercytokinemia.

In a recent study by He et al. [75], the pathological implications of oxidized low-density lipoprotein (ox-LDL) and lectin-like oxLDL receptor-1 (LOX-1) in CAL formation were investigated. Furthermore, the study explored the potential of early plasma ox-LDL concentrations as a predictive factor for CAL in KD. The plasma ox-LDL concentrations and LOX-1 mRNA expression in circulating blood mononuclear cells (MNCs) were significantly higher in patients with KD than in non-KD febrile and healthy children, particularly in the CAL group. The coronary Z-score significantly correlated with plasma oxLDL and LOX-1 mRNA expression. These results suggest that ox-LDL/LOX-1 is involved in CAL development and that the plasma oxLDL concentration in the acute phase is a potentially useful biological indicator for predicting CAL in patients with KD. A summary of biomarkers for oxidative stress at the early stage of KD is provided in Table 1.

## 8. Nitrosative Stress Status in the Acute Stage of KD

In prior clinical investigations, the measurement of nitrite/nitrate (NOx) levels or nitrate alone in blood or urine, both stable byproducts of NO, has served as an indirect means to gauge overall systemic NO production [76]. Iizuka et al. [77] reported that children with KD excreted significantly higher levels of urinary NOx than healthy children (controls) in the initial stage. Patients with CAL excreted significantly higher NOx levels than those without CAL on days 17–21 after fever onset. Tsukahara et al. [78] also reported similar results. In this study, urinary NOx excretion was serially determined in patients with KD. Following IVIG administration, there was a noticeable increase in urinary NOx excretion for each patient, which subsequently decreased to normal levels as clinical and laboratory improvements were observed.

Ikemoto et al. [79] evaluated the serial changes in plasma nitrate levels during the acute stage of KD. The findings revealed a substantial increase in plasma nitrate levels among KD patients from the initial to the third week. Although these levels declined from the third week to the second month, they remained elevated compared to those observed in the healthy control group. Similarly, Wang et al. [80] investigated the plasma and urinary levels of NOx in KD patients both before and three days after IVIG treatment. Initially, patients with KD had significantly higher plasma and urinary levels of NOx than non-KD febrile controls. The elevated plasma NOx levels significantly decreased after IVIG treatment. In contrast, the urinary NOx levels did not change after IVIG treatment. The KD patients with CAL had significantly higher plasma NOx levels than those without CAL before IVIG treatment.

Intriguingly, Wang et al. [80] found that the inducible NOS (NOS2) protein and mRNA were predominantly expressed in MNCs obtained from patients with acute KD but significantly decreased after IVIG treatment. Approximately 95% of the MNCs were positive for the NOS2 protein. The number of NOS2 protein-positive cells decreased by <5% after IVIG treatment. No NOS2 protein expression was detected in MNCs from non-KD febrile control patients, and no NOS1 or NOS3 protein expression was detected in MNCs from KD or febrile control patients. In addition, NOS2 mRNA was significantly higher in MNCs from KD patients than in MNCs from febrile control patients and showed an earlier appearance of 12 PCR cycles, indicating a 2^12^ = 4096-fold increase in NOS2 mRNA expression. After IVIG treatment, NOS2 mRNA expression decreased 2^5^ = 32-fold. No NOS1 or NOS3 mRNA was detected in MNCs from KD or control patients.

Furthermore, Yu et al. [81] determined the cellular origin of NO by conducting a flow cytometric analysis of NOS2 expression in peripheral blood leukocytes and an immunohistochemical analysis of circulating endothelial cells and coronary arteries in children with acute KD. At diagnosis, NOS2 expression was the highest in neutrophils, while its peak in monocytes occurred two weeks after the initial onset. The levels of both cell types were significantly higher in KD patients with CAL than in those without CAL. In addition, the number of circulating endothelial cells and NOS2 expression levels were significantly higher in patients with CAL than in those without. The immunohistochemical analysis of the coronary arteries of three patients with acute KD revealed NOS2 immunoreactivity in endothelial cells and infiltrating monocytes/macrophages in the aneurysms. These results indicate that NO is dynamically generated by neutrophils, monocytes, and endothelial cells at different phases of acute KD and appears to correlate with the severity of coronary arterial wall injury and the progression of CAL in acute KD.

Straface et al. [82] evaluated the contributions of ROS and RNS to systemic oxidative stress in the whole blood of patients with acute KD. Compared to the blood of healthy donors, the blood of patients with KD showed increased oxygen- and nitrogen-derived species production, as detected by using electron paramagnetic resonance spin probing with cyclic hydroxylamine 1-hydroxy-3-carboxy-pyrrolidine. The involvement of the L-arginine-NO pathway was confirmed by decreased concentrations of ADMA (an endogenous NOS inhibitor) and increased concentrations of 3-nitrotyrosine (a product of tyrosine nitration mediated by RNS, such as ONOO^−^ and nitrite) in plasma. Plasma levels of the pro-inflammatory enzyme myeloperoxidase were increased. Circulating erythrocyte alterations typically associated with an oxidative imbalance and premature aging (e.g., decreased total thiol content, glycophorin A, and CD47 expression and increased phosphatidylserine externalization) were also detected. Furthermore, increased platelet activation markers (e.g., degranulation, phosphatidylserine externalization, and leukocyte–erythrocyte–platelet aggregates), decreased antioxidant power, and increased soluble P-selectin and soluble annexin V were found in the blood of patients with acute KD [83]. Since phosphatidylserine-externalizing platelets exert a pro-coagulant activity, these results suggest that the increased risk of vascular complications in KD could depend on platelet stimulation and defective apoptosis, probably related to nitrosative stress.

Yoshimura et al. [84] examined the levels of NO and ROS produced by neutrophils in children with acute KD, using specific fluorescent indicators. This study enrolled children with acute KD, children with non-KD febrile disease, and afebrile controls. Neutrophils from the patients with KD produced significantly higher amounts of NO than those from the patients in the other groups. Concerning ROS, significant increases were found in KD and non-KD febrile patients. In patients with KD, the amount of NO produced by neutrophils decreased after IVIG treatment, whereas ROS production had no significant change.

Finally, Huang et al. [85] determined whether three biomarkers, L-arginine, ADMA, and symmetric dimethylarginine (SDMA), could predict outcomes in children with KD [52,86]. Plasma levels of L-arginine, ADMA, and SDMA were measured in patients with acute KD and in non-KD febrile control patients. Plasma L-arginine, ADMA, and SDMA levels were significantly lower in the patients with KD than in the controls before IVIG treatment (2 g/kg, single dose). After treatment with IVIG, L-arginine, ADMA, L-arginine/ADMA ratios, and arginine methylation ([ADMA + SDMA]/L-arginine) increased significantly. Persistently lower SDMA and higher ADMA/SDMA ratios were observed in the patients with KD than in the control patients. A lower magnitude of change in terms of L-arginine and the ADMA/SDMA ratio after IVIG treatment was associated with CAL formation. These findings suggest that the levels of L-arginine, ADMA, and SDMA are associated with KD and that a reduction in the change in L-arginine and ADMA/SDMA after IVIG treatment is linked to adverse outcomes in acute KD patients. A summary of biomarkers for the L-arginine–NO system at the early stage of KD is provided in Table 2.

As described above, KD occurs with an exacerbated inflammatory response involving the enhanced production of ROS and NO. The excessive in vivo production of ROS and NO (forming ONOO^−^) triggers a vicious spiral of inflammatory reactions and oxidative/nitrosative stress; this presumably leads to diffuse vasculitis during the acute stage of KD. In most cases, systemic inflammation and enhanced oxidative/nitrosative stress can be reduced via the anti-inflammatory activities of IVIG. However, it may be prolonged in some cases of KD; this has recently been identified as a clinical problem in the chronic stage of KD. The next section focuses on this critical issue.

MIS-C is characterized by persistent fever; single or multi-organ dysfunction; and laboratory evidence of inflammation, elevated neutrophils, reduced lymphocytes, and low albumin levels. MIS-C has clinical attributes and cytokine profiles that are comparable to those of KD. Therefore, oxidative/nitrosative stress and endotheliopathy following SARS-CoV-2 infection are thought to contribute to the pathogenesis of MIS-C [87,88,89]. However, relevant data on this topic are lacking, and further research is needed to determine the crosstalk between oxidative/nitrosative stress, endotheliopathy, and systemic inflammation in the pathophysiology of MIS-C.

## 9. Endothelial Dysfunction Syndrome in the Chronic Stage of KD

Patients with CAL in the acute stage of KD may develop aneurysms and are at risk of clinical cardiovascular events and sudden death [6,7,8,9]. Up to 25% of untreated children develop permanent coronary artery damage and subsequent coronary artery aneurysms [6,11,90,91]. “Endothelial dysfunction” is one of the earliest manifestations of arteriosclerosis and has been observed in children with KD with or without CAL after acute illness [68,92,93]. Endothelial dysfunction is characterized by impaired antithrombogenesis, inappropriate vascular smooth muscle tone and proliferation regulation, impaired neutrophil and monocyte adhesion inhibition, and diapedesis. Accumulating evidence indicates that many features of endothelial dysfunction are closely linked to the altered expression and function of the L-arginine-NO pathway and/or enhanced oxidative stress status [43,44,45,46,47,48].

As long as it does not cause fatal complications, such as coronary artery rupture and thromboembolism in the acute stage, systemic diffuse vasculitis can be suppressed with high-dose IVIG anti-inflammatory treatment as the main intervention. Based on the suggestion of Yahata and Hamaoka [68], inflammation can be roughly categorized into three types or groups depending on the degree: (A) CAL that persists, (B) CAL that is transient and regresses at some stage, and (C) CAL that is not detected at all from the acute stage. Regarding Group A, it is widely acknowledged that inflammatory scarring, such as fibrosis, persists over the long term due to vessel remodeling; however, there is no consistent opinion regarding Groups B and C.

Clinical studies in patients who have lived for many years after the resolution of acute KD have been conducted, and a uniform trend has been shown. Vascular function was assessed by measuring arterial flow-mediated vasodilation (FMD) and pulse-wave velocity using high-resolution ultrasonography [94,95,96,97,98,99,100,101,102]. The results of these studies indicate that the possibility of persistent vascular disorders is high in Group B but low in Group C. However, there have been occasional reports of vascular disorders persisting in Group C. Even among the reports with no significant differences compared with healthy controls, some showed tendencies toward a slight decline in vascular function in Group C [68]. Preclinical atherosclerosis may progress predominantly in the coronary artery before progression in other body arteries in patients with severe coronary artery lesions and a history of KD.

Previous studies have provided ample evidence for enhanced oxidative stress, which may reduce NO bioavailability in children after KD. Deng et al. [95] suggested the improvement of arterial endothelial dysfunction in patients (mean age: 7.1 years) with and without coronary aneurysms at 1.0 to 9.6 years after the initial diagnosis of KD via the acute intravenous administration of vitamin C. The authors hypothesized that the antioxidant action of vitamin C might be responsible for its beneficial effects. Cheung et al. [97] reported that, compared with healthy controls, patients with coronary aneurysms (mean age: 13.4 years) had significantly higher serum levels of oxidative stress markers, such as malondialdehyde and THs, and that these markers correlated positively with the carotid intima–media thickness and stiffness index. Ishikawa and Seki [102] measured serum TH levels using a Free Radical Elective Evaluator (FREE^®^; Wismerll Co., Ltd., Tokyo, Japan) [45,67] and compared the levels with brachial arterial FMD in a cohort of 50 children; this encompassed 10 KD patients with CAL (Group 1; median age: 7.5 years), 15 KD patients without CAL (Group 2; median age: 6.7 years), and 25 healthy children matched for age and sex (Group 3). Serum TH levels were significantly higher in Groups 1 and 2 than in Group 3. The %FMD of Groups 1 and 2 was significantly lower than that of Group 3. A significant negative correlation was observed between the serum TH levels and %FMD. These results suggest that oxidative stress is positively associated with endothelial dysfunction in patients with early childhood KD.

Furthermore, Hamaoka et al. [68,99] measured urinary 8-isoprostane, an oxidative stress marker, and urinary NOx, a metabolic product of NO, in patients with KD in the chronic stage. The participants included 149 adolescents and young individuals with a history of KD (mean age: 15.6 years). Urinary 8-isoprostane levels were significantly higher, and NOx levels were significantly lower, regardless of the presence or absence of CAL, in 149 subjects with a history of KD than in 367 control subjects. These results suggest that enhanced oxidative stress is involved in the occurrence and development of vascular endothelial dysfunction in patients with a history of KD, even during the chronic stage.

Hamaoka et al. [99] administered fluvastatin to 11 patients with coronary aneurysms or stenotic lesions for over 12 months. HMG-coenzyme A reductase inhibitors have vascular protective effects because they exert antioxidative effects by activating NOS3 and controlling NADPH. They found a significant improvement in urinary 8-isoprostane and NOx levels and in measures of vascular function such as %FMD and brachial–ankle pulse-wave velocity. Notably, Motoji et al. [103] recently reported that, in their mouse model of KD-like vasculitis, statins show anti-atherosclerotic effects by improving endothelial cell function. 

These results suggest that HMG-coenzyme A reductase inhibitors are useful as alternative therapeutic strategies for stabilizing the continuous post-inflammatory vascular remodeling that results in the development of arteriosclerosis late after KD or the early onset of atherosclerosis in the future.

These findings support the contention that chronic-phase vascular endothelial dysfunction, enhanced oxidative stress, and reduced NO bioavailability are serious clinical problems in children with a history of KD. As these factors closely interact and mutually amplify each other’s effects [43,44,45,46,47,48], they are likely all involved in the onset and progression of arteriosclerotic changes in the chronic stage of KD. Especially when risk factors such as hypertension, diabetes, smoking, obesity, and hyperlipidemia are present, increased oxidative stress and a decreased bioavailability of NO in patients with a history of KD might accelerate atherosclerosis [68].

## 10. Environmental and Toxicological Factors in the Development of KD

### 10.1. Breast Feeding and Risk of KD

KD is an inflammatory disease resulting from abnormal immune responses to possible infectious or environmental stimuli in genetically predisposed individuals [6,7,8,9,14]. Since breastfeeding promotes anti-inflammatory responses and has been shown to protect against both allergic and autoimmune diseases [104,105,106,107,108], it is highly plausible that breastfeeding may also protect against KD.

Indeed, our recent nationwide population-based study of Japanese children that began in 2010 indicated that breastfeeding protects against the development of KD. Yorifuji et al. [109] examined the association between breastfeeding and KD development at 6–30 months of age. We then observed that children who were breastfed exclusively or partially were less likely to be hospitalized for KD than those who were formula-fed without colostrum; the adjusted odds ratio (aOR) for hospitalization was 0.26 (95% confidence interval (CI): 0.12–0.55) for exclusive breastfeeding and 0.27 (95% CI: 0.13–0.55) for partial breastfeeding. Although the risk reduction was not statistically significant, colostrum provided a protective effect. To the best of our knowledge, this is the first study to examine the association between breastfeeding and the development of KD. There are several possible reasons why breastfeeding protects against KD development. First, a mother may provide her immunological memory to her infant through breast milk, preventing the infant from contracting infections that trigger an abnormal immune response. Second, breastfeeding may support the maturation of the immune system, which limits potential damage from an uncontrolled inflammatory response. Furthermore, breast milk is thought to mature the immune system through the establishment of the intestinal microbiota [105,110,111].

Consecutively, we also examined the association between preterm birth and the development of KD from 6 to 66 months of age using data from a nationwide, population-based, longitudinal survey in Japan [112]. We observed that preterm infants were more likely to be hospitalized due to KD (adjusted risk ratio (aRR): 1.55, 95% CI: 1.01–2.39). When we used the combined categories of birth and breastfeeding status, preterm infants who were partially breastfed or formula-fed had a significantly higher risk of hospitalization due to KD than term infants who were exclusively breastfed. Preterm infants are usually born with low immunoglobulin levels. Therefore, an increased susceptibility to infection in preterm infants could persist throughout infancy [113,114]. Infection susceptibility may be associated with an increased risk of developing KD. Given the persistent difference in the gut microbiota between preterm and term infants up to four years of age [105,110,111], an altered gut microbiota in preterm infants could be linked to KD development.

### 10.2. Redox Modulating Factors in Human Breast Milk

The progression of human milk from colostrum to mature milk, facilitated by transitional milk, provides infants with nutrition and safeguards that align with their evolving developmental needs [104,105]. A large body of evidence documents the benefits of human breast milk for infants in reducing morbidity and mortality and protecting against specific infections during breastfeeding. Additional data have demonstrated long-term health benefits for infants (and mothers) beyond lactation. Unlike commercial formulas, human breast milk contains soluble cellular components that give infants passive immunity to the gastrointestinal tract. These antimicrobial and bioactive factors are multifunctional and anti-inflammatory, with established protective roles against necrotizing enterocolitis and other diseases [106,115]. 

Breast milk contains various antioxidative agents, including enzymes (e.g., superoxide dismutase, catalase, and glutathione peroxidase), vitamins (e.g., vitamins A, C, and E and coenzyme Q), reducing agents (e.g., cysteine, glutathione, and thioredoxin), binding proteins (e.g., albumin, ceruloplasmin, and lactoferrin), and constituents of antioxidative enzymes (e.g., zinc, copper, and selenium) [104,107]. In contrast, infant formulas lack significant amounts of antioxidant enzymes or similar agents, making breast milk’s overall antioxidant capacity superior. Shoji et al. [116,117] reported that a sensitive marker of oxidative damage (urinary 8-hydroxy-2-deoxyguanosine) in urine was significantly lower in breastfed term and preterm infants than in formula-fed infants.

Human breast milk contains high concentrations of NOx in the early postpartum period [118,119,120,121,122]. Previously, our group of investigators [120] found high concentrations of NOx (479 ± 274 µmol/L using the Griess method) in 43 milk samples obtained from 32 women during postpartum days 1–8. The highest concentrations of NOx were observed on postpartum days 3–5. Lactating women’s blood contains levels of NOx comparable to those of normal adult blood levels (40.8 ± 27.8 µmol/L). The NOx concentrations in human milk far exceed those found in the blood during the early postpartum period and appear to reach a peak just before the volume of milk increases. These findings indicate that the physiological concentrations of NOx in human breast milk are independent of circulating NOx concentrations and depend mostly on endogenous NO synthesis in the mammary glands [122]. Incidentally, our group [123] also found that the concentrations of thioredoxin, a powerful redox-regulating protein, in early breast milk (268 ± 23 ng/mL during postpartum days 1–8 by using enzyme-linked immunosorbent assays) far exceeded those found in the blood of lactating women (35 ± 7 ng/mL), as well as those found in the blood of healthy adults (20 ± 5 ng/mL).

In a human study conducted by Hord et al. [121], nitrite (NO_2_^−^) concentrations were found to be 0.08 ± 0.02 mg/100 mL (1.7 ± 0.4 µmol/L) in colostrum and 0.001 ± 0.001 mg/100 mL (0.02 ± 0.02 µmol/L) in transition and mature breast milk. This translates to an average daily nitrite intake of 0.2 mg/kg body weight/day in colostrum and 0.0027 mg/kg body weight/day in transition and mature breast milk. These calculations are based on a 3.0 kg body weight and a daily intake volume of 750 mL, which is a commonly used reference volume for infants according to a study by the US Food and Nutrition Board, Institute of Medicine, National Academy of Science (Table 3). Being synchronized with the nitrite content in human milk, both the total antioxidant capacity (measured using a ferric reducing/antioxidant power assay) and free radical scavenging activity (evaluated using 1,1-diphenyl-2-picrylhydrazyl radicals) were found to be significantly higher in colostrum than in transitional and mature milk [124]. The total antioxidant levels showed a trend to decrease from 1062 ± 501 µmol/L in colostrum to 725± 302 µmol/L after 6 months of study. In the 2,2-diphenyl-1-picrylhydrazyl test for radical scavenging activity, the colostrum was more potent (50 ± 20%) in reducing the stable radical 1,1-diphenyl-2-picrylhydrazyl than the transitional and mature milk.

Dietary nitrate/nitrite has beneficial effects on health maintenance and the prevention of lifestyle-related diseases in adulthood by serving as an alternative source of NO through the “enterosalivary nitrate–nitrite–NO pathway” [125,126] (Figure 7). However, this pathway is not as developed during the initial postnatal phase due to the absence of oral commensal nitrate-reducing bacteria and reduced saliva production compared to adults. Colostrum boasts the highest nitrite content compared to transitional, mature, and even artificial milk to counterbalance the reduction in nitrite levels in this timeframe; this implies that colostrum is vital in replenishing nitrite until the enterosalivary nitrate–nitrite–NO pathway becomes established in the subsequent postnatal period. Therefore, nitrite-rich breast milk can prevent neonatal infections and other diseases, including KD [115,121]. 

### 10.3. Ambient Pollution and Risk of KD

Recent reviews have shown a causal association between prenatal smoking and asthma incidence [127,128], and accumulated evidence supports the contention that prenatal exposure to particulate matter can induce immune dysregulation, leading to a higher occurrence of KD. Using data from a nationwide, population-based, longitudinal survey in Japan, Yorifuji et al. [129] examined the effects of prenatal and postnatal exposure to particulate matter on KD hospital admissions from 6 to 30 months of age. Children exposed to higher levels of suspended particulate matter, particularly during pregnancy, were more likely to be hospitalized for KD. The aOR values for ≥25 µg/m^3^ exposure compared with <20 µg/m^3^ exposure were 1.59 (95% CI: 1.06–2.38) for prenatal exposure and 1.41 (95% CI: 0.82–2.41) for postnatal exposure. Therefore, prenatal exposure during mid-to-late gestation increases the risk of KD.

Yorifuji et al. [130] also examined the association between early childhood exposure to maternal smoking and the incidence of KD. Maternal smoking status was ascertained at six months of age, and responses to questions about hospital admissions for KD between 6 and 30 months were used as the outcomes. Maternal smoking increased the risk of admission, particularly between 6 and 18 months of age, in a dose-dependent manner. Compared with children of non-smoking mothers, the children of mothers who smoked had an aRR of 1.83 (95% CI: 1.06–3.35) for hospital admissions between 6 and 30 months of age and an aRR of 2.69 (95% CI: 1.56–4.64) for hospital admissions between 6 and 18 months of age. Therefore, early childhood exposure to maternal smoking may increase KD’s hospitalization risk.

Furthermore, Yorifuji et al. [131] examined the association between neonatal sepsis and the further development of KD using data from the National Hospital Organization Neonatal Intensive Care Unit registry study in Japan. A multivariate logistic regression analysis adjusted for preterm birth, sex, the use of antibiotics in the neonatal intensive care unit, parity, and maternal smoking showed that children with neonatal sepsis were more likely to have a history of KD at three years of age (aOR: 11.67, 95% CI: 2.84–47.96). Although there are few reports on subsequent immunological follow-ups in infants with neonatal sepsis, a previous study showed that children with a history of neonatal sepsis had lower IgE levels and fewer asthma complications in childhood [132]. Our results suggest that neonatal sepsis could have a long-term effect on immune status. Among infants admitted to the neonatal intensive care unit, sepsis during hospitalization may be associated with the risk of developing KD later in life. Perinatal infection control is important from the perspective of long-term complications. Although oxidative stress mechanisms in sepsis are highly complicated, ROS and RNS play pivotal roles in its evolution [133]. Neonatal sepsis triggers innate immune and inflammatory responses, leading to systemic oxidative stress and tissue damage.

As described in the preceding sections, oxidative and nitrosative stresses play critical roles in the pathophysiology of inflammation-based KD [7,8,9]. The excessive in vivo production of ROS and RNS contributes to a never-ending vicious spiral of hyperinflammatory reactions. Pro-oxidant environments associated with factors such as tobacco smoke exposure, air pollution, and sepsis may underlie the development of acute-stage KD. Therefore, further exploration of this field and additional research are necessary to mitigate the morbidity and mortality associated with KD.

## 11. Adjunct Therapy for Vascular Endothelial Dysfunction in Patients with KD

KD and COVID-19 have similar vascular pathologies [134,135]. The amelioration of endothelial dysfunction may prevent or delay the development of arteriosclerotic processes. However, there is insufficient evidence indicating that antioxidant treatments improve vascular endothelial function in these diseases.

Statins are therapeutic agents used to treat hypercholesterolemia by inhibiting HMG-CoA reductase, an enzyme involved in the cholesterol biosynthetic pathway. They also have cardiovascular protective effects that are independent of their cholesterol-lowering properties, known as “pleiotropic” effects. Statins have been shown to suppress the activity of pro-oxidant enzymes (such as NADPH oxidase) and pro-inflammatory transcriptional pathways in the endothelium. By enhancing endothelial NOS (NOS3) expression and activity, they exert antioxidant effects and are thought to contribute to the reduction in cardiovascular tissue damage caused by oxidative and nirosative stress [136,137]. 

As described in Section 9, Hamaoka et al. [68,99] administered fluvastatin to 11 KD patients with coronary aneurysms or stenotic lesions (eight males and three females; age at onset: 1–3 years; age at the time of the study: 7–25 years) for 12 months. They found a significant improvement in urinary 8-isoprostane and NOx levels, serum high-sensitivity C-reactive protein levels, and measures of vascular function such as %FMD and brachial–ankle pulse-wave velocity during treatment. The dose of fluvastatin (0.5–0.7 mg/kg/day) was controlled so that the total serum cholesterol level was maintained above 120 mg/kg in each subject. No adverse events were reported. These results suggest that HMG-coenzyme A reductase inhibitors are a promising therapeutic strategy for stabilizing the continuous post-inflammatory vascular remodeling that results in the development of arteriosclerosis in the chronic stage of KD. 

Recently, Tremoulet et al. [138] performed a Phase 1/2a, dose-escalation study of atorvastatin (0.125–0.75 mg/kg/day) in 34 patients with acute KD (age: 2–17 years) and echocardiographic evidence of coronary artery aneurysms. They showed that a six-week course of atorvastatin at doses up to 0.75 mg/kg/day was well tolerated with no major drug-related adverse events. Considering the potential benefit of the anti-inflammatory and immunomodulatory actions of atorvastatin, a Phase 3 efficacy trial will be warranted in the near future.

More recently, Shimizu, et al. [139] examined genome-wide transcriptional changes in cultured human umbilical vein endothelial cells incubated with sera from patients with KD treated with or without the addition of atorvastatin to standard therapy. Those patients had been recruited in the above-described trial [138]. The endothelial cells exposed to sera from patients with acute KD who received standard therapy plus atorvastatin exhibited gene expression profiles indicative of improved endothelial cell health and reduced inflammation compared to the control group. Specifically, the sera from patients with acute KD treated with atorvastatin blocked the deleterious effect of pretreatment sera on cultured endothelial cells, characterized by a reduced expression of NOS3 and an increased expression of IL-8. These findings may have implications for the use of atorvastatin to preserve endothelial cell homeostasis in children with acute vascular inflammation attributable to KD.

There are no reports on the efficacy of statins in MIS-C; however, the clinical efficacy of statins in adult patients with COVID-19 has been reported. Several recent studies have shown that statin supplementation reduces the requirement for invasive ventilation in patients with severe COVID-19 [140,141]. In other studies, statin use before hospitalization for COVID-19 was found to be associated with a reduction in the advancement of severity and in-hospital mortality [142,143,144]. Randomized controlled trials are needed to validate these results.

Vitamin C is a potent water-soluble antioxidant and an effective scavenger of free radicals, such as superoxide anion (O_2_^−^•), which could improve endothelial dysfunction by increasing the availability of NO [145]. Deng et al. [95] evaluated whether the acute administration of vitamin C could improve endothelial dysfunction using the percent change in the diameter of the brachial artery induced by reactive hyperemia. They showed that systemic endothelial dysfunction develops after KD, even after early treatment with IVIG, but that it could be restored after the acute administration of vitamin C. They suggested that the underlying mechanism was endothelin-1, which could increase superoxide anion (O_2_^−^•) production.

The other therapeutic options for endothelial dysfunction in the chronic stage of KD include thiol compounds (N-acetylcysteine, carbocysteine, and erdosteine) and Nrf2 activators (sulforaphane, bardoxolone methyl, and dimethyl fumarate). Plant-derived polyphenols (resveratrol, quercetin, and curcumin) and dietary antioxidants (vitamin C and vitamin E) may also improve and support the antioxidant defense mechanisms in the body [46]. 

NO-stimulating dietary supplements (i.e., “NO boosters”) are widely available and have shown promise as safe therapeutic options to improve endogenous NO regulation in cardiovascular diseases associated with endothelial dysfunction [125]. While L-arginine serves as the precursor for endothelial NOS (NOS3)-mediated NO synthesis, oral L-arginine supplementation is often ineffective in significantly enhancing NO synthesis or bioavailability. This ineffectiveness can be attributed to several factors [46,49,50]. L-citrulline, found in high concentrations in watermelon, is a neutral amino acid formed by enzymes in the mitochondria that also serves as a substrate for recycling L-arginine. In contrast to L-arginine, L-citrulline is not extensively extracted by the gastrointestinal tract (specifically enterocytes) or the liver. Consequently, supplementing with L-citrulline is more effective in raising L-arginine levels and promoting endogenous NO synthesis [146,147]. 

The above-mentioned pharmacological and nutritional agents are relatively well tolerated and might be beneficial for the clinical management of the chronic stage of KD. It is expected that future studies may reveal pharmaceutical and bioactive compounds that are actually protective for vascular health in KD.

## 12. Concluding Remarks

KD is characterized by systemic vasculitis accompanied by immunoregulatory abnormalities and is a common cause of childhood-acquired heart disease in most developed countries. The cause of KD remains elusive, but existing data suggest that “endothelial dysfunction” plays a pivotal role in the development of arteriosclerosis in this condition. Many of the features of endothelial dysfunction are intimately linked to oxidative stress enhancement and L-arginine–NO system derangements.

The impact of NO on cell function or death is complex and often appears to be contradictory. NO may be cytotoxic but may also protect cells from a toxic insult; it is an antioxidant but may also compromise the cellular redox state via the oxidation of thiols, such as glutathione. The situation may be even more complicated, as NO may react with oxygen or superoxide anion (O_2_^−^•) to produce nitrogen reactive species with a much broader chemical reaction spectrum than NO itself. Thus, the action of NO during inflammatory reactions should be considered in the context of the degree, duration, and timing of its synthesis.

Enhanced oxidative stress and decreased NO bioavailability play crucial roles as both promoters and mediators of vascular inflammation and arteriosclerotic changes, suggesting potential targets for the treatment of KD in the chronic phase. Unfortunately, there have been relatively few successes in the clinical setting, and the experiences with statins and “NO boosters” are quite limited. Before these therapies can be widely adopted in clinical practice, comprehensive longitudinal studies must be carried out to assess panels of oxidative and nitrosative stress biomarkers in conjunction with traditional clinical endpoints in KD patients. The results would contribute toward a better understanding of the role played by NO in the pathogenesis and therapeutic options of KD.

## Figures and Tables

**Figure 1 ijms-24-15450-f001:**
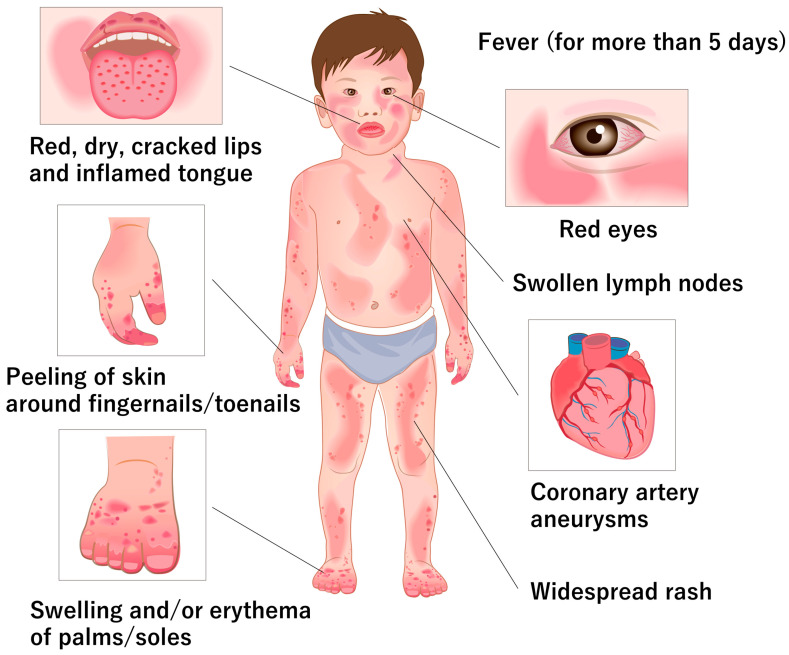
Major clinical features of Kawasaki disease.

**Figure 2 ijms-24-15450-f002:**
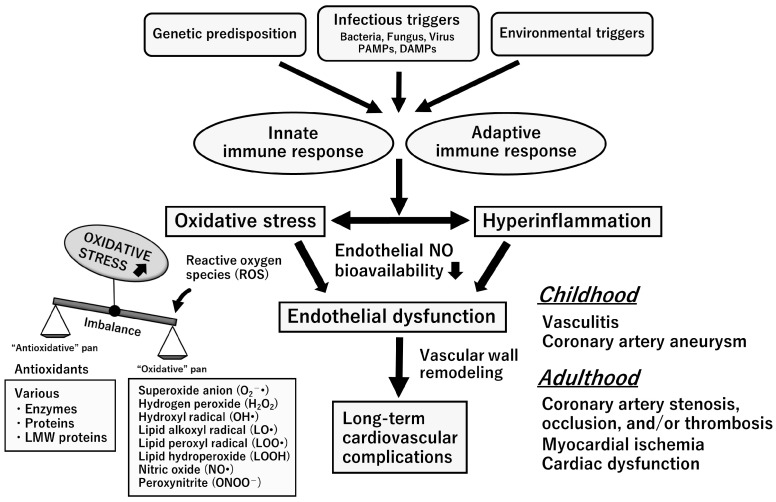
Etiology and pathophysiology of vascular endothelial dysfunction in Kawasaki disease. Kawasaki disease is triggered by various factors, such as genetic predisposition and infectious and environmental triggers. Oxidative stress and systemic inflammation are induced in the acute phase through both the innate immune response and adaptive immune response. These lead to endothelial dysfunction and long-term cardiovascular complications, such as coronary artery stenosis, occlusion, and/or thrombosis. Abbreviations: DAMPs, damage-associated molecular patterns; LMW, low molecular weight; PAMPs, pathogen-associated molecular patterns.

**Figure 3 ijms-24-15450-f003:**
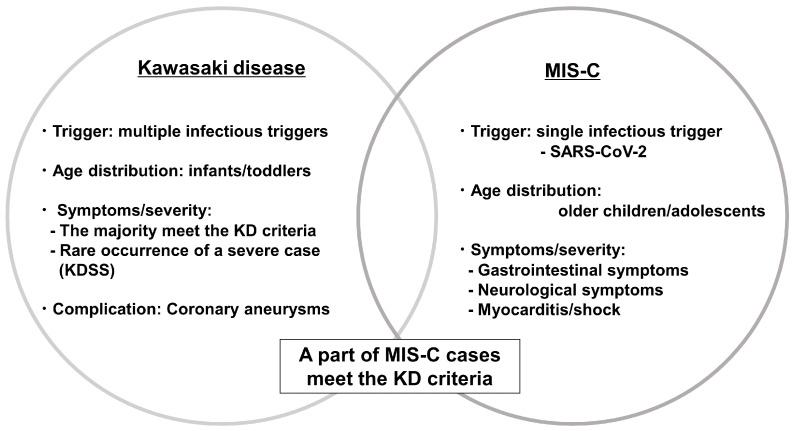
Kawasaki disease and multisystem inflammatory syndrome in children (MIS-C). Abbreviations: KD, Kawasaki disease; KDSS, Kawasaki disease shock syndrome; SARS-CoV-2, severe acute respiratory syndrome coronavirus 2.

**Figure 4 ijms-24-15450-f004:**
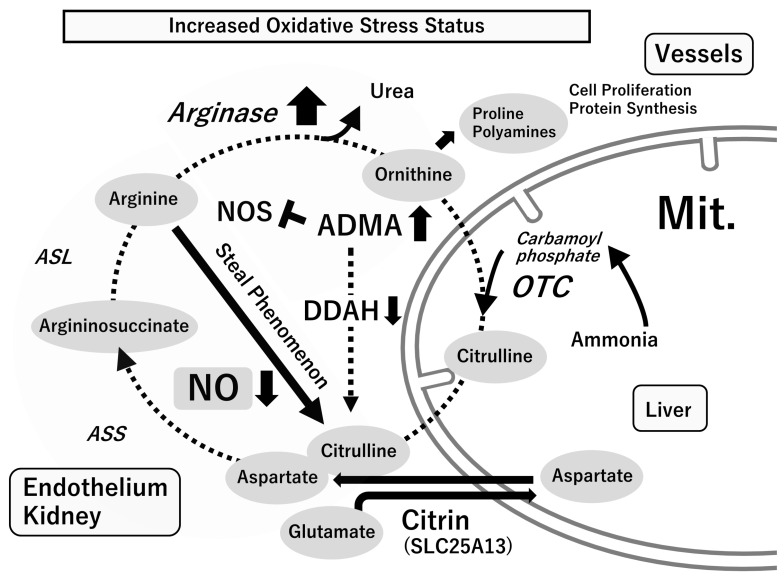
Metabolic network centered on L-arginine. In the physiological state, arginine is regenerated from citrulline via “citrulline–arginine recycling” in the blood vessel and the kidney, and the vascular endothelial cells can maintain NO synthesis via this pathway. In inflammatory pathologies (increased oxidative stress status), arginase is induced in many cells, such as the blood vessels and macrophages, and NO synthesis derived from the vascular endothelium decreases through the “arginine steal phenomenon”. The increased catabolism of arginine via arginase may not only compromise the ability to synthesize NO constitutively but may also increase the production of ornithine, a precursor for the synthesis of proline and polyamines required for cell proliferation and collagen synthesis. ADMA, asymmetric dimethylarginine; ASL, argininosuccinate lyase; ASS, argininosuccinate synthetase; DDAH, dimethylarginine dimethylaminohydrolase; NO, nitric oxide; NOS, nitric oxide synthase; OTC, ornithine transcarbamylase. Reprinted by permission from the journal “*Antioxidants*”.

**Figure 5 ijms-24-15450-f005:**
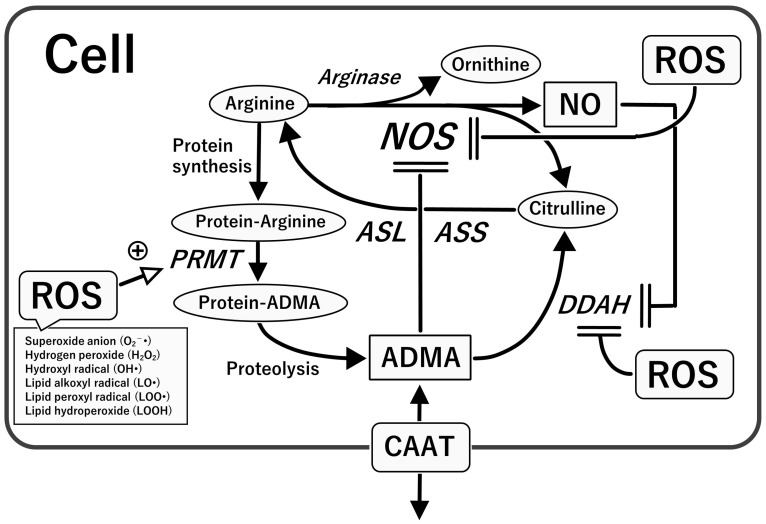
L-arginine-NOS-NO pathway versus PRMT-ADMA-DDAH pathway. Note that ROS enhance the activity of PRMT and that the action of DDAH is suppressed by ROS (or high levels of NO). Notably, endothelial NOS (NOS3) is suppressed by ROS (or high levels of NO). Thus, oxidative stress enhancement and endothelial NOS inhibition play pivotal roles in cardiovascular pathologies by regulating PRMT/DDAH activity and NO synthesis, leading to “endothelial dysfunction”. ADMA, asymmetric dimethylarginine; ASL, argininosuccinate lyase; ASS, argininosuccinate synthase; CAAT, cationic amino acid transporter; DDAH, dimethylarginine dimethylaminohydrolase; NO, nitric oxide; NOS, nitric oxide synthase; PRMT, protein arginine methyltransferase; ROS, reactive oxygen species.

**Figure 6 ijms-24-15450-f006:**
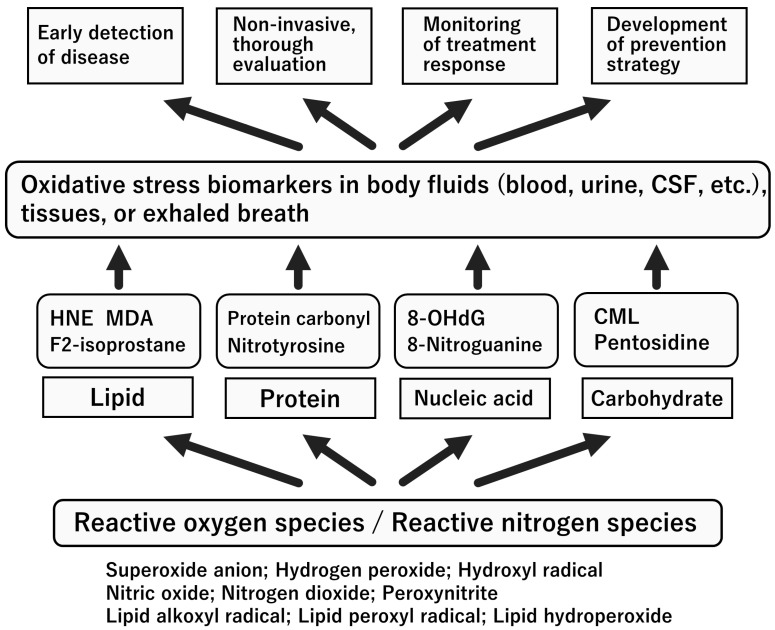
Perspective on the clinical use of oxidative stress biomarkers. Abbreviations: CML, carboxymethyl lysine; CSF, cerebrospinal fluid; HNE, 4-hydroxy-2-nonenal; MDA, malondialdehyde; 8-OHdG, 8-hydroxy-2′-deoxyguanosine. Only the representative markers are presented here.

**Figure 7 ijms-24-15450-f007:**
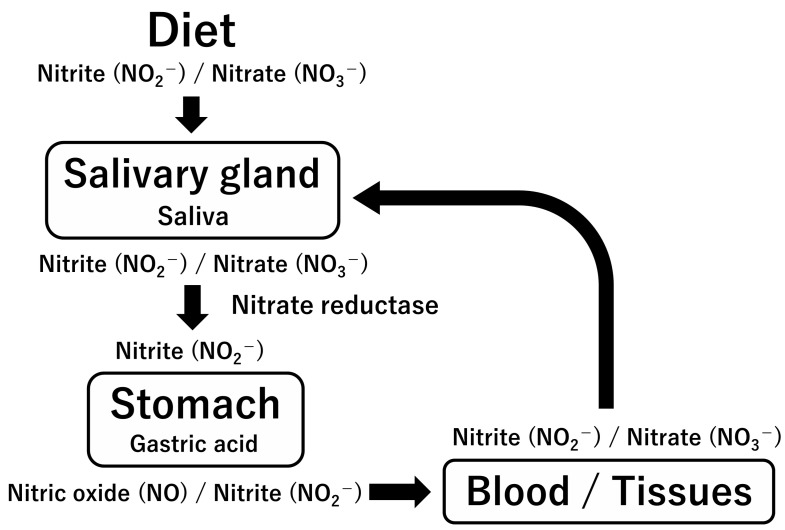
Enterosalivary nitrate-nitrite-NO pathway.

**Table 1 ijms-24-15450-t001:** Biomarkers for oxidative stress status at the early stage of Kawasaki disease (before specific treatments).

Formation of Modified Molecules by Reactive Oxygen Species:	Kawasaki Disease vs. Controls
Lipid peroxides (plasma)	↑ [69]
Malonyldialdehyde/organic hydroperoxides (plasma)	↑ [70]
Total hydroperoxides (blood)	↑ [71] ^a^
Total hydroperoxides (serum)	↑ [72] ^a^
8-isoprostane (urine)	↑ [73]
Oxidized low-density lipoprotein (plasma)	↑ [75]
Lectin-like oxidized low-density lipoprotein receptor-1 (mononuclear cells)	↑ [75]
Antioxidative molecules and enzymes:	
Manganese superoxide dismutase (leukocytes)	↑ [69]
Glutathione peroxidase/catalase (erythrocytes)	↑ [69]
Copper/zinc superoxide dismutase (leukocytes/erythrocytes)	→ [69]
Heme oxygenase-1 (monocytes)	↑ [74]
Others:	
Zinc (plasma)	↓ [70]
Biological antioxidative potential (blood)	→ [71] (IVIG-responding) ^a^ ↓ [71] (IVIG-nonresponding) ^a^
Biological antioxidative potential (serum)	↓ [72] ^a^

↑: increased levels; ↓: reduced levels; →: no difference. Notably, sex-related differences were not examined in any of these studies. ^a^ These markers were measured using a free radical elective evaluator (Wismerll Co., Ltd., Tokyo, Japan). The “oxidative stress index”, which is the ratio of total hydroperoxides to biological antioxidative potential, seems to be a more sensitive oxidative stress biomarker than individual markers [72]. Abbreviation: IVIG, intravenous immunoglobulin.

**Table 2 ijms-24-15450-t002:** Biomarkers for L-arginine–nitric oxide system at the early stage of Kawasaki disease (before specific treatments).

Biomarkers of L-Arginine–Nitric Oxide System	Kawasaki Disease vs. Controls
Nitrite/nitrate (urine)	↑ [77] ^a^
Nitrite/nitrate (urine)	↑ [78]
Nitrate (plasma)	↑ [79]
Nitrite/nitrate (plasma/urine)	↑ [80]
Inducible nitric oxide synthase (NOS2) (mononuclear cells)	↑ [80] ^a^
Inducible nitric oxide synthase (NOS2) (neutrophils/circulating endothelial cells)	↑ [81]
Inducible nitric oxide synthase (NOS2) (monocytes)	→ [81] ^b^
Reactive oxygen- and nitrogen-derived species (e.g., peroxynitrite) (whole blood)	↑ [82] ^c^
Asymmetric dimethylarginine (plasma)	↓ [82]
Nitric oxide (neutrophils)	↑ [84] ^d^
L-arginine/asymmetric dimethylarginine/symmetric dimethylarginine (plasma)	↓ [85]

↑: increased levels; ↓: reduced levels; →: no difference. Notably, sex-related differences were not examined in any of these studies. ^a^ NOS1 and NOS3 expression was not detected in mononuclear cells from the patients with Kawasaki disease [80]. ^b^ NOS2 expression in monocytes showed an increase at two weeks after disease onset, and this expression persisted at the four-week evaluation [81]. ^c^ Reactive oxygen- and nitrogen-derived species were detected using electron paramagnetic resonance spin probing with cyclic hydroxylamine 1-hydroxy-3-carboxy-pyrrolidine [82]. ^d^ Intracellular nitric oxide level was measured using diaminofluorescein-FM diacetate as a specific fluorescent indicator [84]. Abbreviation: NOS, nitric oxide synthase.

**Table 3 ijms-24-15450-t003:** Nitrate and nitrite concentrations in human milk and daily consumption [115].

Stage	Nitrate (in 100 mL)	Nitrite (in 100 mL)	Nitrate (in 750 mL)	Nitrite (in 750 mL)
Colostrum	0.19 ± 0.03 (mg) 3.1 ± 0.5 (µmol)	0.08 ± 0.02 (mg) 1.7 ± 0.4 (µmol)	1.43 ± 0.24 (mg) 23.1 ± 3.9 (µmol)	0.60 ± 0.15 (mg) 13.0 ± 3.3 (µmol)
Transition	0.52 ± 0.10 (mg) 8.4 ± 1.6 (µmol)	0.001 ± 0.001 (mg) 0.02 ± 0.02 (µmol)	3.90 ± 0.75 (mg) 62.9 ± 12.1 (µmol)	0.008 ± 0.008 (mg) 0.2 ± 0.2 (µmol)
Mature	0.31 ± 0.02 (mg) 5.0 ± 0.3 (µmol)	0.001 ± 0.001 (mg) 0.02 ± 0.02 (µmol)	2.32 ± 0.21 (mg) 37.4 ± 3.4 (µmol)	0.008 ± 0.008 (mg) 0.2 ± 0.2 (µmol)

The daily intake of nitrate and nitrite was calculated based on a daily intake volume of 750 mL.

## Data Availability

Not applicable.

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
