# Peer review of "Roles of Oxidative Injury and Nitric Oxide System Derangements in Kawasaki Disease Pathogenesis: A Systematic Review"

_ijms, 2023, doi:10.3390/ijms242015450_

Round 1
Reviewer 1 Report

Needs some English language editing.
Author Response
To Reviewer 1:
We are grateful to Reviewer 1 for the critical comments and useful suggestions that have helped us improve our paper. As indicated in the responses that follow, we have considered all these comments and suggestions while revising our manuscript.
Point 1: Figure 2 is inaccurate and should be corrected. The concept that KD leads to atherosclerosis is not correct and the figure is misleading. In Japan where rates of smoking are high, known risk factors for atherosclerosis may be superimposed on vascular injury from KD, but this is NOT a direct link. Autopsy data from the U.S. demonstrates that young adults can die from coronary artery thrombosis with virtually no atherosclerotic changes. (Orenstein et al. 2012, Shimizu et al. 2015, Lee et al. 2015). In the legend, delete : “These lead to endothelial dysfunction and long-term complications such as arteriosclerosis and atherosclerosis.” In addition, the term “cytokine storm” should be eliminated. This phrase was coined for MAS, HLH, and applied to MIS-C and COVID-19. The cytokine levels in KD are much lower compared to these hyperinflammatory states and the term cytokine storm should not be used for KD.
Response 1: Thank you for your valuable comment. We have deleted the descriptions of "cytokine storm," "arteriosclerosis," and "atherosclerosis" in Figure 2. We also removed the sentence “These lead to endothelial dysfunction and long-term complications such as arteriosclerosis and atherosclerosis.” from the legend (Line 108-114).
Point 2: Line 144: MIS-C does NOT lead to coronary artery aneurysms. All references to this should be removed in the paper. The published cases of “CAA in MIS-C” are likely misdiagnosed KD and KD shock. The pathophysiology of MIS-C does not support vasculitis. Figure 3: remove aneurysms from MIS-C.
Response 2: Thank you for bringing this to our attention. We have deleted the description and references of coronary artery aneurysm as a complication of MIS-C (Line 154-155, Figure 3).
Point 3: Line 565: Would be appropriate to cite the phase I/IIa trial of atorvastatin by Tremoulet et al. and the HUVEC expts by Shimizu et al. that examine the effects of atorvastatin on KD patients.
Response 3: Your valuable comment is greatly appreciated. We added two references about atorvastatin and a description about them according to your suggestions in the paragraph of “adjunct therapy for vascular endothelial dysfunction in patients with KD” (Line 802-820).
Point 4: Line 72: Using Z score criteria, approx.. 40% of patients will have a Z max >2.5, which is classified as an aneurysm. See Ogata et al. Int J Cardiol 2013.
Response 4: Thank you for your valuable comment. We have added the description as you pointed out (Line 81-84).
Point 5: Line 149: Person-to-person transmission of a KD agent was supported by the decrease in KD cases worldwide associated with mitigation strategies to decrease the spread of COVID-19: masking, school closures, social distancing. This should be stated more clearly. See Burney et al. JAMA Network 2022, and papers from Korea, Western Europe.
Response 5: Thank you for bringing this to our attention. We have added the description as you pointed out. (Line 168-175).
Point 6: Line 571: KD does NOT lead to accelerated atherosclerosis in the absence of classic athero risk factors.
Response 6: Your valuable comment is greatly appreciated. We have revised the text to avoid the misunderstanding that KD directly leads to accelerate atherosclerosis: “Especially when risk factors such as hypertension, diabetes, smoking, obesity, and hyperlipidemia are present, increased oxidative stress and decreased bioavailability of NO in patients with a history of KD might accelerate arteriosclerosis” (Line 626-629)
Point 7: Needs some English language editing.
Response 7: Thank you for your advice. The English editing was performed by native speaking proofreaders again.
Reviewer 2 Report
This review is well articulated and documented.
For both KD and MIS-C, the authors should verify and add data on gender differences in ROS/RNS production, susceptibility to vascular injury, and development of cardiovascular events.
Straface and coworkers hypothesized that in KD nitrosative stress may activate platelets and increase the risk of vascular complications. Based on these data the authors, in addition to red blood cells, should also mention platelets in the text. (Straface et al., Biochem Biophys Res Commun. 2010)
Author Response
To Reviewer 2:
We are grateful to Reviewer 2 for the critical comments and useful suggestions that have helped us improve our paper. As indicated in the responses that follow, we have considered all these comments and suggestions while revising our manuscript.
Point 1: For both KD and MIS-C, the authors should verify and add data on gender differences in ROS/RNS production, susceptibility to vascular injury, and development of cardiovascular events.
Response 1: Thank you for the important suggestion. We added a description of the sex ratio in KD and MIS-C patients, and we added the descriptions that male is a risk factor for coronary artery lesions in KD patients and that cardiac complications were observed more common in male patients than in female patients (Line 158-164). Unfortunately, no reports were found that investigated sex differences in ROS/RNS production in KD and MIS-C patients.
Point 2: Straface and coworkers hypothesized that in KD nitrosative stress may activate platelets and increase the risk of vascular complications. Based on these data the authors, in addition to red blood cells, should also mention platelets in the text (Straface et al., Biochem Biophys Res Commun. 2010).
Response 2: We thank the Reviewer for this insightful suggestion. We added the description about the increased risk of vascular complications in KD which is dependent on platelet stimulation associated with nitrosative stress (Line 496-502).
Reviewer 3 Report
Reviewer report
Oxidative injury and nitric oxide system derangements in the pathogenesis of Kawasaki disease.
This was a very well-written and comprehensive paper that focuses light on the various aspects of oxidative injury and nitric oxide system in the Kawasaki disease-affected children.
1. Can authors add related literature in the form of table which will be helpful to readers?
2. Please add subheadings after each heading in the article to divide the content in a systematic way.
3. line 35 what is the global statistics?
4. Line 53 What are the environmental factors please list out.
5. Line 91 PAMPS and DAMPS include a number of agents make it clearer which are causative agents for KD.
6. Can authors add a graphical representation of the figure instead of texts format. e.g. figures can be made from Bio-render or any Online available software.
English language writing is fine.
Author Response
To Reviewer 3:
We are grateful to Reviewer 3 for the critical comments and useful suggestions that have helped us improve our paper. As indicated in the responses that follow, we have considered all these comments and suggestions while revising our manuscript.
Point 1: Can authors add related literature in the form of table which will be helpful to readers?
Response 1: Thank you for the important suggestion. We have added a table listing literature on biomarkers of oxidative stress status (Table 1) and a table listing literature on L-arginine-nitric oxide system biomarkers before treatment for Kawasaki disease (Table 2).
Point 2: Please add subheadings after each heading in the article to divide the content in a systematic way.
Response 2: We thank the Reviewer for this insightful suggestion. We added subheadings after some headings in the manuscript (Lines 191, 247, 631, 666, and 733).
Point 3: Line 35 what is the global statistics?
Response 3: Thank you for your suggestion. We have added a description of the prevalence of Kawasaki disease worldwide (Lines 35-40).
Point 4: Line 53 What are the environmental factors please list out.
Response 4: We appreciate your attention to this matter. We had already included a description in the manuscript that listed environmental (Line 105-106); therefore, we didn't add a new description.
Point 5: Line 91 PAMPs and DAMPs include a number of agents make it clearer which are causative agents for KD.
Response 5: Thank you for pointing this out. We added a description of related PAMPs and DAMPs as causative agents for KD (Lines 101-104).
Point 6: Can authors add a graphical representation of the figure instead of texts format. e.g. figures can be made from Bio-render or any Online available software.
Response 6: Thank you for your advice. We agree that the figures created using the software you suggested will be much easier for readers to understand. Figures in our manuscript are not a complicated figure; therefore, we believe that the current figures are sufficient for readers to understand without the modification. We are so grateful for your advice again.
Round 2
Reviewer 1 Report
The authors have responded clearly to suggestions for editing. A few comments follow:
1) Line 83: “have Z-max scores ≥2.5, which are classified as coronary aneurysms” Please add the word “initial” before Z-max scores because it is important to point out that the 40% figure refers to the first echo before treatment is completed.
2) Line 563: “High-dose IVIG treatment is the primary intervention for suppressing systemic dif- fuse vasculitis in the acute stage, provided it does not lead to fatal complications such as 564 coronary artery rupture and thromboembolism”. Suggest re-wording this sentence as it sounds like IVIG could lead to fatal complications.
3) Lines 627-630: The authors should use the term “atherosclerosis” , not arteriosclerosis, referring to the accumulation of lipids and cholesterol clefts within the vascular wall.
Author Response
To Reviewer 1:
We are grateful to Reviewer 1 for the critical comments and useful suggestions that have helped us improve our paper. As indicated in the responses that follow, we have considered all these comments and suggestions while revising our manuscript.
Point 1: Line 83: “have Z-max scores ≥2.5, which are classified as coronary aneurysms” Please add the word “initial” before Z-max scores because it is important to point out that the 40% figure refers to the first echo before treatment is completed.
Response 1: Thank you for the important suggestion. As you pointed out, we added "initial" before Z-max scores (Line 83-84).
Point 2: Line 563: “High-dose IVIG treatment is the primary intervention for suppressing systemic diffuse vasculitis in the acute stage, provided it does not lead to fatal complications such as coronary artery rupture and thromboembolism”. Suggest re-wording this sentence as it sounds like IVIG could lead to fatal complications.
Response 2: We thank the Reviewer for this insightful suggestion. To avoid misunderstandings among readers, we have changed the description as follows (Line 562-564).
“As long as it does not cause fatal complications, such as coronary artery rupture and thromboembolism in the acute stage, systemic diffuse vasculitis can be suppressed with high-dose IVIG anti-inflammatory treatment as the main intervention.”
Point 3: Lines 627-630: The authors should use the term “atherosclerosis” , not arteriosclerosis, referring to the accumulation of lipids and cholesterol clefts within the vascular wall.
Response 3: Thank you for your advice. We changed "arteriosclerosis" to "atherosclerosis" (Line 629).